# When Planning Fails Despite Correct Execution:
# On Epistemic Calibration for LLM-Based Multi-Agent Systems

**Zehao Wang** [1 2]   **Shilong Jin** [1 2]   **Zhao Cao** [† 3]   **Lanjun Wang** [† 2]

## Abstract

LLM-based multi-agent systems can fail even when planned actions are executed correctly because agents may misjudge their knowledge when evaluating plan feasibility, a phenomenon we term epistemic miscalibration in planning. Unlike execution errors, epistemic miscalibration is latent during planning, as generated plans can remain self-consistent and executable without observable errors; the miscalibration is also dynamic, as new information can alter feasibility assessments, potentially obscuring past miscalibration signals and causing them to recur over time. To address this, we propose the Epistemic Planning Calibration Agentic Workflow (EPC-AW), which assesses whether plans remain supported under varying information conditions rather than directly verifying feasibility. EPC-AW employs Information-consistency-based Plan Selection, selecting plans whose evaluations are stable across agents, together with Consistency-guided Epistemic State Refinement to adapt calibration over time by leveraging past discrepancies to guide future planning. Experiments show that EPC-AW improves system-level success by an average of 9.75%. Code is available in the public repository (https://github.com/wzhSteve/EPC-AW).

## 1. Introduction

Large language model-based multi-agent systems have become a dominant paradigm for complex decision making, tool use, and long-horizon task execution (Becattini et al., 2025; Sun et al., 2025; He et al., 2025). By decomposing

tasks across interacting agents, such systems enable scalable problem solving beyond the capability of single-agent approaches (Becattini et al., 2025; Ronanki, 2025). However, despite their rapid development, these multi-agent systems remain fragile in practice, with failures frequently observed in real deployments (Cemri et al., 2025; Hammond et al., 2025). Diagnosing and correcting such failures has therefore become an emerging requirement for reliable agentic systems (Liang et al., 2025; Epperson et al., 2025).

Recently, a growing body of work has sought to repair failures in LLM-based multi-agent systems by intervening at different stages of system operation, including post-hoc and online methods (Liang et al., 2025; Ma et al., 2025). Post-hoc approaches (Epperson et al., 2025; Ma et al., 2025) analyze interaction traces after task failure and apply retrospective corrections, while online methods (Liang et al., 2025; Shen et al., 2025) monitor error accumulation during task progression and intervene to prevent errors from escalating into task failure. Across these approaches, failures are predominantly attributed to execution-level faults, such as logically incorrect tool outputs or invalid tool returns (Zhang et al., 2025b; Lu et al., 2025; Reid et al., 2025).

However, practical deployments reveal a distinct class of failures that cannot be explained by execution faults alone. Even when all actions are correctly executed following the prescribed plan, the task may remain unsolved because the plan itself is infeasible with respect to the intended objective. Such failures arise at the planning stage, where the plan encodes a mismatch between intended goals and the actions required to achieve them. For example, a plan may invoke external tools and receive valid responses, yet fail to acquire verifiable evidence for the objective. This occurs when the plan is based on a miscalibrated feasibility assessment, causing the system to repeatedly execute valid actions without ever satisfying the goal.

We characterize this failure mode as *epistemic miscalibration in planning*: a planning agent assigns unwarranted confidence to feasibility assessments, failing to recognize the limits of its knowledge in tasks requiring iterative information acquisition. This phenomenon poses two key challenges. First, epistemic miscalibration is latent. Unlike execution errors, which manifest as observable errors in action real-

[1]College of Intelligence and Computing, Tianjin University, Tianjin, China [2]School of New Media and Communication, Tianjin University, Tianjin, China [3]Gaoling School of Artificial Intelligence, Renmin University of China, Beijing, China. Correspondence to: Zhao Cao <caozhao@ruc.edu.cn>, Lanjun Wang <wanglanjun@tju.edu.cn>.

*Proceedings of the 43rd International Conference on Machine Learning*, Seoul, South Korea. PMLR 306, 2026. Copyright 2026 by the author(s).

ization or inconsistencies in reasoning traces (Zhang et al., 2025c; Cemri et al., 2025), epistemic miscalibration arises from erroneous feasibility assessments that leave no explicit error signals. Consequently, plans affected by epistemic miscalibration can appear coherent and executable, making such failures particularly difficult to detect at the planning stage. Second, epistemic miscalibration is dynamic. As the planning agent continuously acquires new information, its feasibility assessments evolve, which can gradually obscure past miscalibration signals and lead to repeated miscalibration over time. While prior methods leverage dynamic feedback to correct execution-level errors (Ma et al., 2025; Epperson et al., 2025), they rely on observable errors as supervision and therefore do not address plan-level epistemic miscalibration, limiting their ability to sustain calibration under evolving information.

To mitigate epistemic miscalibration, we propose the *Epistemic Planning Calibration Agentic Workflow* (EPC-AW), a planning-centered workflow for multi-agent collaboration. EPC-AW does not attempt to directly verify plan feasibility. Instead, it focuses on whether the plan continues to be supported by the agents under different information conditions. EPC-AW comprises two complementary components. *Information-consistency-based Plan Selection* (IPS) operates within each round and serves as a planning-time diagnostic. Rather than checking whether the plan is judged as feasible, IPS examines whether a plan's evaluation remains stable across agents that possess different information. Plans whose evaluation varies substantially across heterogeneous information conditions are treated as epistemically fragile, while plans with stable evaluations are selected. *Consistency-guided Epistemic State Refinement* (CESR) operates across rounds by recording discrepancies between the planning agent's locally selected plans and the IPS-selected plans, interpreting these discrepancies as signals for epistemic miscalibration. These signals are integrated into persistent memory, which constrains subsequent planning and prevents previously observed miscalibration patterns from reoccurring as information accumulates. Together, EPC-AW shifts failure mitigation from execution-time correction to planning-time epistemic calibration. In summary, this paper makes the following contributions:

- We formalize *epistemic miscalibration in planning* as a repair target in LLM-based multi-agent systems, revealing failures arising from miscalibrated planning assessments even under correct execution.
- To address epistemic miscalibration in planning assessments under the absence of observable errors, we propose IPS, which selects plans whose evaluations remain stable across agents operating under heterogeneous information.
- To adapt epistemic calibration under evolving information, we introduce CESR, a memory-driven mechanism

that leverages past calibration errors to constrain future planning and suppress persistent misjudgments.
- Extensive experiments on six LLM-based multi-agent benchmarks demonstrate that EPC-AW significantly improves task success, achieving an average 9.75% increase in system-level success rate.

## 2. Related Works

### 2.1. Failure and Repair in LLM-based Multi-Agent Systems

LLM-based multi-agent systems have been increasingly adopted for complex reasoning, tool use, and long-horizon decision making (Becattini et al., 2025; Ronanki, 2025; Li et al., 2024), but they remain vulnerable to failures caused by reasoning errors, coordination breakdowns, and unreliable information use (Dobrovsky et al., 2025; Hammond et al., 2025; Zhang et al., 2025a). To improve system reliability, prior work has proposed mechanisms for diagnosing and repairing failures by analyzing execution traces and agent interactions during or after runtime (Cemri et al., 2026; Shen et al., 2025; Epperson et al., 2025; Ma et al., 2025).

Existing approaches differ in when and how corrections are applied. Some methods perform post-hoc or cross-run debugging by inspecting interaction traces or validating failure hypotheses through repeated executions (Epperson et al., 2025; Ma et al., 2025). Others introduce online monitoring and correction, such as rollback and reflection during execution (Liang et al., 2025) or history-conditioned anomaly detection with local repairs (Shen et al., 2025). Despite these differences, most methods focus on correcting incorrect actions or local reasoning errors during execution.

In contrast, this work formulates *epistemic miscalibration in planning* as a distinct repair target in LLM-based multi-agent systems. It reveals that system-level failures can arise from miscalibrated feasibility assessments at planning time, even when all subsequent executions are locally correct.

### 2.2. Model- and Agent-Level Epistemic Calibration

Epistemic calibration in large language models has been widely studied in the context of overconfidence, and uncertainty estimation (Abbasi Yadkori et al., 2024; Lee et al., 2025; Chhikara, 2025). This line of work treats miscalibration as a property of an individual model or agent, aiming to align expressed confidence with actual knowledge.

Several methods leverage multi-agent structures to enhance calibration through consensus, voting, or verifier agents that critique intermediate outputs (Clark et al., 2025; Wen et al., 2024; Pitre et al., 2025). These approaches rely on LLMs acting as judges, but such first-order judgments are themselves subject to epistemic miscalibration, limiting their

ability to diagnose epistemic failures. Another related line of work draws from peer prediction and Bayesian truth inference (Witkowski & Parkes, 2012; Chen et al., 2025), which calibrate reports using incentive signals and repeated feedback under static game-theoretic assumptions.

In contrast, we study epistemic miscalibration as a system-level failure mode in running LLM-based multi-agent systems. Our goal is to repair task failures by intervening during system operation, where there is no additional payoff signals or external supervision. Motivated by peer-based calibration methods, we exploit the stability of agents' evaluations across heterogeneous information and refine epistemic states over time based on persistent cross-agent inconsistencies. Further details are provided in Appendix. G.

## 3. Problem Formulation

### 3.1. Operation of LLM-based Multi-Agent Systems

An LLM-based multi-agent system is denoted by $\mathcal{M}$ and consists of multiple interacting agents that coordinate through structured messages, shared memory, and external tools (Li et al., 2026; Wu et al., 2024). Given a user query $Q$, the system aims to produce an answer by iteratively planning, executing tool-mediated actions, and aggregating newly acquired information.

At iteration $t$, the system maintains an information context $\mathcal{I}^{(t)}$, which aggregates the user query, previously acquired evidence, and interaction history. Conditioned on $\mathcal{I}^{(t)}$, a planning agent generates a plan $\pi^{(t)} = (g^{(t)}, a^{(t)})$, where $g^{(t)}$ denotes an intermediate goal and $a^{(t)}$ specifies an action intended to acquire information relevant to $g^{(t)}$ through available tools, such as web search or code execution. The selected action is executed, yielding an observation or piece of evidence $e^{(t)}$, which is incorporated into the information context via an aggregation operator

$$\mathcal{I}^{(t+1)} = \mathrm{D}(\mathcal{I}^{(t)}, \pi^{(t)}, e^{(t)}), \tag{1}$$

where $\mathrm{D}(\cdot)$ integrates new evidence into the existing context.

This planning-execution loop continues until a stopping criterion is met. The overall behavior of the system defines a mapping from the user query to a final answer,

$$\hat{Y} = \mathcal{M}(Q), \tag{2}$$

where $\hat{Y}$ denotes the generated answer to the query $Q$.

### 3.2. Epistemic Miscalibration in Planning

*Epistemic miscalibration in planning* refers to a failure of feasibility assessment, in which the agent assigns unwarranted confidence to its judgments and fails to recognize the limits of its knowledge in tasks that require iterative information acquisition.

Let $\phi$ denote the latent factual situation of the task, which specifies, in principle, the complete information required to justify a plan. This information is not directly observable and must be acquired through iterative interactions.

Given the information available at step $t$, denoted by $\mathcal{I}^{(t)}$, the planning agent implicitly affirms the feasibility of a plan $\pi^{(t)}$. The subjective feasibility assessment is formulated as

$$J(\pi^{(t)} \mid \mathcal{I}^{(t)}). \tag{3}$$

where $J(\cdot)$ denotes an abstract judgment function and this assessment reflects the agent's confidence that executing the action $a^{(t)}$ in the plan can satisfy the current goal $g^{(t)}$ through external tool invocations under the available information context $\mathcal{I}^{(t)}$.

In contrast, the objective feasibility of a plan $\pi^{(t)}$ under the latent factual situation $\phi$ is characterized by

$$E(\pi^{(t)} \mid \phi), \tag{4}$$

which indicates whether the plan can, in principle, be justified given the complete underlying information.

Epistemic miscalibration in planning arises when the agent's feasibility assessment formed under partial information, $J(\pi^{(t)} \mid \mathcal{I}^{(t)})$, is misaligned with the objective feasibility condition $E(\pi^{(t)} \mid \phi)$ defined under complete information.

### 3.3. Problem Statement

Given a user query $Q$, an LLM-based multi-agent system produces a final answer $\hat{Y}$, whose correctness is evaluated against the ground-truth answer $Y^\star$. In practice, even all executions are correct, failures often arise from epistemic miscalibration during the planning phase.

The problem addressed in this work is to design an agentic workflow that mitigates epistemic miscalibration in planning by guiding the system toward plans whose feasibility assessments remain well-aligned with the information available as the system acquires new evidence. By improving epistemic calibration at the planning stage, the workflow aims to reduce downstream errors and increase the likelihood that the final output satisfies $\hat{Y} = Y^\star$.

## 4. EPC-AW: Epistemic Planning Calibration Agentic Workflow

Epistemic miscalibration originates at planning and thus persists even under correct execution and valid tool outputs. Moreover, continual information updates shift feasibility assessments, allowing miscalibration to recur in the subsequent planning. To address these challenges, we propose the *Epistemic Planning Calibration Agentic Workflow* (EPC-AW), which intervenes directly at planning time by separating planning, execution, and diagnosis under heterogeneous

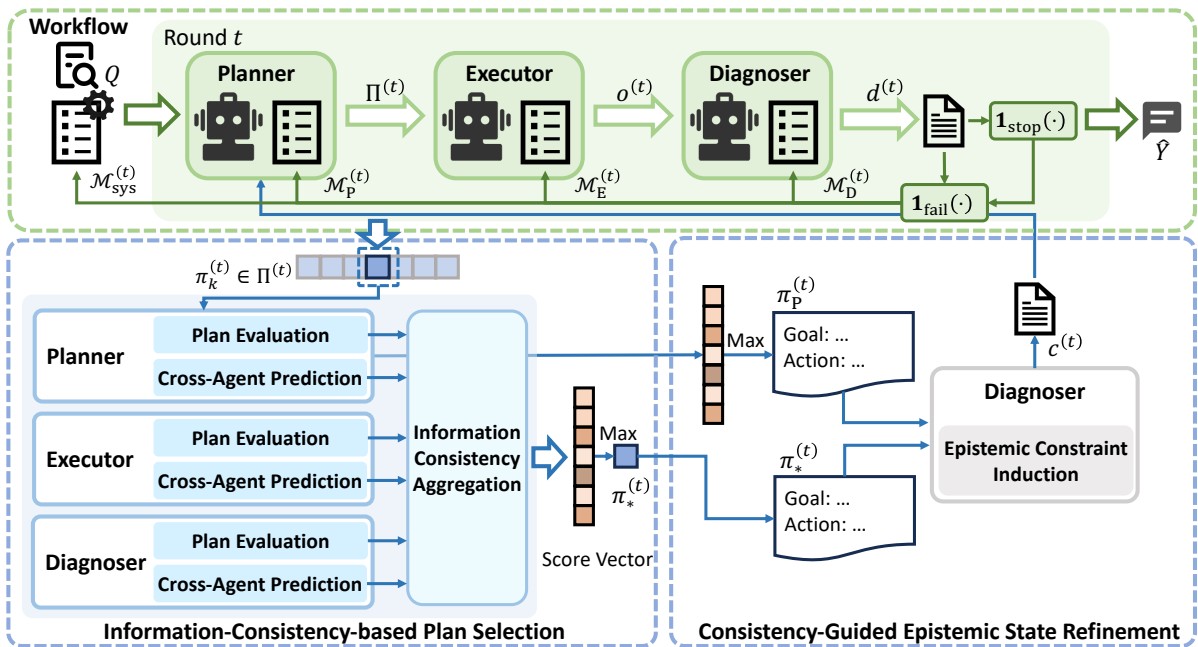

*Figure 1.* Overview of EPC-AW. EPC-AW consists of three agents, the Planner, Executor, and Diagnoser, each with heterogeneous information in memory. At each round, Information-consistency-based Plan Selection evaluates candidate plans across agents and selects those with stable evaluations, providing a planning-time calibration signal. Across rounds, Consistency-guided Epistemic State Refinement aggregates consistency feedback to guide planning under evolving information. The process terminates upon satisfying a stopping condition, after which the final answer is produced.

information. The overview is shown in Fig. 1, with architectural details in Sec. 4.1. EPC-AW operates both within rounds and across rounds. Within each round, *Information-consistency-based Plan Selection* (IPS) favors plans whose evaluation remain stable across agents with heterogeneous information, mitigating epistemic miscalibration in planning assessments in the absence of observable error signals (Sec.4.2). Across rounds, *Consistency-guided Epistemic State Refinement* (CESR) aggregates consistency feedback into persistent constraints that guide subsequent planning as information evolves (Sec.4.3).

### 4.1. System Architecture and Workflow

EPC-AW is instantiated as a role-specialized LLM-based multi-agent system consisting of three agents with fixed roles: a *Planner* ($A_P$), an *Executor* ($A_E$), and a *Diagnoser* ($A_D$). The system operates over discrete interaction rounds $t = 1, 2, \ldots, T$. Within each round, agent roles and communication interfaces remain unchanged, forming a stabilized interaction protocol.

**Workflow.** At round $t$, the Planner $A_P$ generates an intermediate plan $\pi^{(t)} = (g^{(t)}, a^{(t)})$, where $g^{(t)}$ denotes the goal and $a^{(t)}$ denotes the action intended to acquire evidence relevant to $g^{(t)}$ through external tools. The Executor $A_E$ instantiates $a^{(t)}$ by invoking the corresponding tool

with concrete inputs and parameters, producing an execution outcome $o^{(t)}$. The Diagnoser $A_D$ evaluates whether the outcome $o^{(t)}$ meets the plan $\pi^{(t)}$, producing diagnostic feedback $d^{(t)}$ that is communicated back to the Planner.

**Memory Structure.** EPC-AW maintains a shared *system-level memory* and *agent-level memories* that are private to individual agents. The agent-level memories induce heterogeneous epistemic states across agent roles.

At round $t$, the complete interaction history is denoted as $\mathcal{H}^{(t)} = \{(\pi^{(k)}, o^{(k)}, d^{(k)})\}_{k=1}^{t-1}$, where $\pi^{(k)}$ is the planned action, $o^{(k)}$ the execution outcome, and $d^{(k)}$ the diagnostic evaluation. This complete history is not directly accessible to any single agent. To formalize the execution analysis within $d^{(k)}$, we define a diagnostic indicator as

$$\mathbf{1}_{\text{fail}}(d^{(k)}) = \begin{cases} 1, & \text{if } d^{(k)} = \text{UNSUPPORTED}, \\ 0, & \text{otherwise.} \end{cases} \quad (5)$$

where $d^{(k)} = \text{UNSUPPORTED}$ indicates that the execution outcome $o^{(k)}$ does not meet the plan $\pi^{(k)}$.

**System-Level Memory.** The system-level memory at round $t$ is defined as

$$\mathcal{M}_{\text{sys}}^{(t)} = \langle Q, \ \Psi^{(t)}, \mathcal{U} \rangle, \quad (6)$$

where $Q$ denotes the user query, $\Psi^{(t)}$ is a set of verifiable evidence accumulated over rounds, and $\mathcal{U} = \{U_P, U_E, U_D\}$ denotes the abstract description of the role for all agents.

At round $k$, an evidence item $\psi^{(k)}$ is extracted from the execution outcome $o^{(k)}$ only if the execution outcome meets the plan. Accordingly, the system-level memory is updated in a set-augmentation manner:

$$\Psi^{(k)} = \Psi^{(k-1)} \cup \{\psi^{(k)}\} \quad \text{if } \mathbf{1}_{\text{fail}}(d^{(k)}) = 0, \quad (7)$$

**Agent-level Memory.** Each agent $\mathcal{A}^i \in \{A_P, A_E, A_D\}$ maintains a private agent-level memory $\mathcal{M}_i^{(t)}$, defined as a role-specific projection of $\mathcal{H}^{(t)}$. The Executor maintains a complete execution trace $\mathcal{M}_E^{(t)} = \mathcal{H}^{(t)}$.

In contrast, the Planner and Diagnoser apply complementary selection criteria. The Planner records plans that are not satisfied by the executions:

$$\mathcal{M}_P^{(t)} = \{(\pi^{(k)}, o^{(k)}, d^{(k)}) \in \mathcal{H}^{(t)} \mid \mathbf{1}_{\text{fail}}(d^{(k)}) = 1\}. \quad (8)$$

Conversely, the Diagnoser selectively records executions whose outcomes satisfy the corresponding plan:

$$\mathcal{M}_D^{(t)} = \{(\pi^{(k)}, o^{(k)}, d^{(k)}) \in \mathcal{H}^{(t)} \mid \mathbf{1}_{\text{fail}}(d^{(k)}) = 0\}. \quad (9)$$

**Information State.** Each agent $A_i$ operates under a private information state

$$\gamma_i^{(t)} \triangleq (U_i, \mathcal{M}_i^{(t)}), \quad (10)$$

which induces heterogeneous information encodings even under a shared system-level memory $\mathcal{M}_{\text{sys}}^{(t)}$.

## 4.2. Information-consistency-based Plan Selection

In realistic multi-agent planning scenarios, the feasibility of a plan is typically not verifiable at the planning phase. Moreover, epistemic miscalibration does not manifest as explicit execution errors or observable reasoning failures. To address this challenge, we introduce *Information-consistency-based Plan Selection* (IPS). Rather than attempting to only assess feasibility, IPS uses cross-agent evaluation consistency as a planning-time signal of epistemic support. It prioritizes plans whose evaluations remain stable across agents operating under heterogeneous information states.

### 4.2.1. PLANNER CANDIDATE PLAN GENERATION

At round $t$, the Planner $A_P$ generates a finite set of candidate plans conditioned on the shared system-level memory $\mathcal{M}_{\text{sys}}^{(t)}$ and its private information state $\gamma_P^{(t)}$:

$$\Pi^{(t)} = A_P\left(\mathcal{M}_{\text{sys}}^{(t)}, \gamma_P^{(t)}\right), \quad (11)$$

where $\Pi^{(t)} = \{\pi_1^{(t)}, \ldots, \pi_K^{(t)}\}$, and each $\pi_k^{(t)} \in \Pi^{(t)}$ represents a structurally distinct strategy.

### 4.2.2. INFORMATION-CONDITIONED PLAN EVALUATION

A common evaluation interface maps information states and candidate plans to real-valued scores:

$$\mathcal{E} : (\gamma_i^{(t)}, \mathcal{M}_{\text{sys}}^{(t)}, \Pi^{(t)}) \longrightarrow \mathbb{R}^K. \quad (12)$$

Each agent computes its evaluation vector

$$\mathbf{e}_i^{(t)} = \mathcal{E}(\gamma_i^{(t)}, \mathcal{M}_{\text{sys}}^{(t)}, \Pi^{(t)}), \quad (13)$$

where $e_i^{(t)}(k) \in \mathbf{e}_i^{(t)}$ reflects agent $i$'s assessment of plan $\pi_k^{(t)}$ under its information state.

### 4.2.3. CROSS-AGENT EVALUATION PREDICTION

Under epistemic miscalibration, an agent's direct evaluation is biased by its private information, making self-assigned scores an unreliable indicator of feasibility. We therefore leverage cross-agent evaluations under heterogeneous information as signals. Each agent predicts how candidate plans would be evaluated under alternative information states, and plans whose predicted evaluations remain stable across these perspectives are less sensitive to epistemic variation and thus less prone to epistemic miscalibration.

For each agent $j \neq i$, agent $i$ constructs an approximation of $j$'s information state

$$\hat{\gamma}_{i \to j}^{(t)} \triangleq (U_j, \mathcal{M}_i^{(t)}). \quad (14)$$

Then, agent $i$ applies the evaluation interface to obtain prediction scores

$$\hat{e}_{i \to j}^{(t)}(k) = \mathcal{E}\left(\hat{\gamma}_{i \to j}^{(t)}, \mathcal{M}_{\text{sys}}^{(t)}, \Pi^{(t)}\right)_k. \quad (15)$$

Collecting these predictions defines the cross-agent evaluation vector $\widehat{\mathbf{e}}_{i \to -i}^{(t)} = \{\hat{e}_{i \to j}^{(t)}(k)\}_{j \neq i, \, k \in [K]}$.

Agent $i$ summarizes the predicted evaluations by averaging over peers:

$$\bar{e}_{-i}^{(t)}(k) = \frac{1}{|\mathcal{A}| - 1} \sum_{j \neq i} \hat{e}_{i \to j}^{(t)}(k), \quad (16)$$

which represents agent $i$'s estimate of the plan's expected evaluation under heterogeneous information.

### 4.2.4. INFORMATION-CONSISTENCY AGGREGATION

To quantify cross-agent information consistency, each agent compares its own plan evaluation with the simulated aggregate evaluation from other agents. The agent-level information-consistency score is defined as

$$s_i^{(t)}(k) = \log\left(e_i^{(t)}(k)\right) - \log\left(\bar{e}_{-i}^{(t)}(k)\right). \quad (17)$$

Aggregating across all agents yields the plan-level information-consistency score:

$$C_{\text{IPS}}(\pi_k^{(t)}) = \frac{1}{|\mathcal{A}|} \sum_{i \in \mathcal{A}} s_i^{(t)}(k). \qquad (18)$$

Plans with higher $C_{\text{IPS}}(\pi_k^{(t)})$ indicate that the plan remains more consistently favored across heterogeneous information-conditioned evaluations, serving as a heuristic measure of cross-agent information consistency under heterogeneous information states. Then, the IPS-selected plan $\pi_*^{(t)}$ at round $t$ is formulated as

$$\pi_*^{(t)} = \arg \max_{\pi_k^{(t)} \in \Pi^{(t)}} C_{\text{IPS}}(\pi_k^{(t)}). \qquad (19)$$

### 4.3. Consistency-guided Epistemic State Refinement

While Information-consistency-based Plan Selection (IPS) calibrates planning decisions within a single round, it cannot prevent epistemic miscalibration from recurring over long-horizon interactions. As planning proceeds, the Planner's internal memory is continually updated, inducing non-stationary plan generation behavior that per-round selection alone cannot correct. We therefore introduce *Consistency-guided Epistemic State Refinement* (CESR), a memory-driven mechanism that accumulates cross-round consistency signals and integrates them into persistent epistemic state constraints, shaping subsequent planning and preventing the reintroduction of previously identified miscalibration.

#### 4.3.1. CROSS-ROUND PLAN DIVERGENCE

Within round $t$, two plans are distinguished. The Planner selects a plan based on its self-conditioned evaluation,

$$\pi_P^{(t)} = \arg \max_{\pi_k^{(t)} \in \Pi^{(t)}} e_P^{(t)}(k), \qquad (20)$$

while IPS selects the information-consistent plan $\pi_*^{(t)}$. A divergence $\pi_P^{(t)} \neq \pi_*^{(t)}$ serves as a diagnostic signal that the Planner's selected plan is more susceptible to epistemic miscalibration than the plan whose evaluations remain stable under heterogeneous information.

#### 4.3.2. EPISTEMIC CONSTRAINT INDUCTION

Upon detecting such a discrepancy, the Diagnoser generates a lightweight epistemic constraint

$$c^{(t)} = \mathcal{G}\left(\pi_P^{(t)}, \pi_*^{(t)}, \mathcal{M}_{\text{sys}}^{(t)}\right), \qquad (21)$$

which abstracts the salient information and structural features associated with the potential epistemic miscalibration revealed by the discrepancy.

The constraint is accumulated in the Planner's memory via

$$\mathcal{M}_P^{(t+1)} = \mathcal{M}_P^{(t)} \cup \{c^{(t)}\}. \qquad (22)$$

### 4.4. Answer Generation and Termination Control

At each round, the Diagnoser determines whether the accumulated system-level information suffices to answer the user query $Q$. This decision is captured by a binary indicator and we formalize it as follows:

$$\mathbf{1}_{\text{stop}}^{(t)}\left(Q, \Psi^{(t)}\right) = \begin{cases} 1, & \text{if sufficient evidence,} \\ 0, & \text{otherwise.} \end{cases} \qquad (23)$$

If $\mathbf{1}$, it transitions to answer generation; otherwise, the system proceeds to the next round.

Upon termination, the final answer is produced by

$$\hat{Y} = \mathcal{G}\left(Q, \Psi^{(t)}, \mathcal{H}^{(t)}\right), \qquad (24)$$

where $\Psi^{(t)}$ contains accumulated verified evidence and $\mathcal{H}^{(t)}$ summarizes the interaction history.

## 5. Experiment

### 5.1. Experimental Setup

#### 5.1.1. DATASET

We evaluate on six benchmarks covering diverse reasoning and search demands. **Bamboogle** (Press et al., 2023) targets compositional multi-step reasoning with minimal external search. **2Wiki** (Ho et al., 2020) emphasizes multi-hop question answering with strong dependence on information retrieval across heterogeneous sources. **HotpotQA** (Yang et al., 2018) requires multi-hop reasoning over Wikipedia with moderate search complexity. **Musique** (Trivedi et al., 2022) stresses sequential reasoning with tightly coupled intermediate inferences. **GAIA** (Mialon et al., 2023) evaluates agentic planning in open-world settings, heavily relying on search and tool use. **MedQA** (Yang et al., 2024) focuses on clinical reasoning without external retrieval.

Overall, these datasets span a spectrum from reasoning-intensive inference to search-driven open-world problem solving, enabling systematic evaluation of epistemic miscalibration across diverse planning and reasoning settings. More details are provided in Appendix A.

#### 5.1.2. BASELINES

We compare EPC-AW with three baselines implemented under the AgentFlow framework (Li et al., 2026), which differ in how epistemic miscalibration is handled during the planning phase. *No-Repair* follows the original Agent-Flow setting, generating and executing a single forward plan without any diagnosis or recovery. *Retry*, inspired by metacognitive retry strategies in MAST (Shen et al., 2025), detects epistemic miscalibration at the current planning step and re-generates the plan locally using diagnostic feedback.

*Table 1.* Accuracy (%) comparison of different repair methods. Δ denotes the absolute improvement of EPC-AW over the corresponding baseline method on each benchmark. Darker gray indicates larger improvement.

| Method | Bamboogle | | 2Wiki | | HotpotQA | | Musique | | GAIA | | MedQA | |
|---|---|---|---|---|---|---|---|---|---|---|---|---|
| | Acc. | Δ | Acc. | Δ | Acc. | Δ | Acc. | Δ | Acc. | Δ | Acc. | Δ |
| No-Repair (AgentFlow) | 48.27 | ↑9.86 | 39.00 | ↑12.83 | 48.33 | ↑12.67 | 9.33 | ↑8.00 | 7.09 | ↑8.13 | 67.11 | ↑7.00 |
| Retry | 51.47 | ↑6.66 | 42.83 | ↑9.00 | 51.33 | ↑9.67 | 10.67 | ↑6.66 | 9.17 | ↑6.05 | 71.00 | ↑3.11 |
| Rollback | 52.27 | ↑5.86 | 44.83 | ↑7.00 | 56.33 | ↑4.67 | 14.00 | ↑3.33 | 10.76 | ↑4.46 | 71.56 | ↑2.55 |
| EPC-AW | **58.13** | - | **51.83** | - | **61.00** | - | **17.33** | - | **15.22** | - | **74.11** | - |

*Rollback*, inspired by COCO (Liang et al., 2025), performs system-level recovery by rolling back the entire system state to a selected historical step before re-planning. All baselines share the same agent architecture, system memory, and execution history, and restrict miscalibration diagnosis to the planning phase for fair comparison. Additional details are provided in Appendix. B.

### 5.1.3. IMPLEMENTATION

All agents in EPC-AW are instantiated using Qwen3-Coder-30B (Yang et al., 2025). The model is deployed on a server equipped with four NVIDIA RTX 4090 GPUs and served via vLLM for efficient inference. During plan generation, the Planner samples $n = 9$ candidate next-step plans with temperature $0.9$ to encourage exploration, while all other generations operate at temperature $0$ for deterministic execution. Candidate plans are evaluated using a predefined feasibility metric, yielding scores from 1 to 5. The maximum number of iterations is set to 10. In IPS, plan evaluations and predictions are performed independently for each agent, with outputs obtained via separate LLM calls.

The system interacts with five tools following Agent-Flow (Li et al., 2026): a base generator, a Python coder, a Google Search, a Wikipedia Search, and a Web Search. To reduce external dependencies, we replace AgentFlow's official Google Search API calls with a lightweight local pipeline that retrieves the top-10 search results via a Chrome-based interface. To further mitigate information leakage due to publicly available datasets, we apply keyword-based filtering during the search stage to exclude HuggingFace repositories. Retrieved pages are then processed by the Web Search tool, followed by LLM-based summarization.

For evaluation, GPT-4o is employed as an automatic judge to determine whether model predictions match the corresponding ground-truth answers, following standard practice in tool-augmented reasoning benchmarks (Li et al., 2026; Arif et al., 2024). To reduce the impact of stochasticity, all experiments are repeated three times, and we report the average accuracy across runs. All evaluation settings are aligned with those used in AgentFlow to facilitate direct comparison. Additional details are provided in Appendix. C.

### 5.2. Performance Analysis

Table 1 summarizes the accuracy of all methods across six benchmarks. All repair-based methods consistently outperform the No-Repair baseline, indicating that addressing epistemic miscalibration mitigates task failures. On average, Retry improves accuracy from 36.52% to 39.41%, while Rollback further increases performance to 41.63% (+5.11% over No-Repair). EPC-AW achieves the best results on all benchmarks, yielding a 9.75% absolute improvement over No-Repair and a 4.64% gain over Rollback.

Comparing the two baselines, Retry performs local re-planning at the current step, whereas Rollback enables recovery from earlier planning decisions by reverting the system state. This broader intervention scope allows Rollback to consistently outperform Retry across all benchmarks, with especially notable improvements on HotpotQA (+5.00%) and Musique (+3.33%).

However, both Retry and Rollback rely on agent-generated diagnosis and repair signals. Since the agents themselves remain epistemically miscalibrated, the resulting failure analysis and recovery strategies can still be unreliable. Consequently, although these methods partially alleviate execution errors, they do not directly reduce the epistemic uncertainty underlying erroneous plan selection.

By contrast, EPC-AW intervenes directly during planning. Through posterior aggregation in IPS and persistent corrective constraints in CESR, EPC-AW reduces epistemic risk during plan selection and prevents recurrent miscalibration across rounds, resulting in substantially stronger and more consistent performance gains.

### 5.3. Ablation Study

We conduct ablation studies on four benchmarks spanning distinct reasoning and search characteristics. GAIA and 2Wiki represent search-dominant settings, with GAIA emphasizing open-world exploration and tool use, and 2Wiki focusing on retrieval-intensive multi-hop QA, whereas Bamboogle targets compositional reasoning with minimal search and HotpotQA occupies an intermediate reasoning and retrieval regime. The remaining benchmarks are omitted due

to substantial overlap with the selected set.

To isolate component contributions, we evaluate two controlled variants. EPC-AW[†] removes both IPS and CESR, approximating planning without epistemic repair with agents under heterogeneous information, while EPC-AW[‡] retains IPS but disables CESR, corresponding to plan selection without constraint feedback.

*Table 2.* Ablation study on four representative benchmarks, evaluating the individual and combined effects of IPS and CESR.

| Method | Bamboogle | 2Wiki | HotpotQA | GAIA |
|---|---|---|---|---|
| EPC-AW[†] | 45.87 | 40.00 | 51.33 | 10.24 |
| EPC-AW[‡] | 46.67 | 36.83 | 49.00 | 10.24 |
| EPC-AW | **58.13** | **51.83** | **61.00** | **15.22** |

As shown in Table 2, IPS alone exhibits regime-dependent effects. On the reasoning-centric Bamboogle benchmark, EPC-AW[‡] yields modest gains, indicating that IPS can regularize planning by favoring epistemically supported reasoning paths. However, on search-intensive benchmarks such as 2Wiki, the same variant degrades performance, suggesting that IPS in isolation induces overly conservative planning that suppresses necessary exploration.

CESR plays a critical corrective role by transforming this conservatism into structured planning guidance. By identifying sources of epistemic miscalibration through plan preference discrepancies and enforcing memory-based constraints across rounds, CESR enables the Planner to operate within safer feasibility regions without suppressing exploration. As a result, the full EPC-AW model consistently outperforms both ablations across all datasets, achieving gains of up to +15.00% on 2Wiki and +12.00% on HotpotQA, indicating that intra-round plan selection and cross-round constraints are jointly essential for robust agentic planning.

### 5.4. Mean-Score Aggregation vs. Cross-Agent Consistency

We compare Information-consistency-based Plan Selection (IPS) with a score aggregation baseline, denoted as Mean-Score Aggregation (MSA), which selects plans by averaging feasibility scores from three agents operating under heterogeneous information. This comparison isolates cross-agent consistency from naive score aggregation, highlighting their distinct effects in agentic planning.

*Table 3.* Comparison between EPC-AW with Mean-Score Aggregation (MSA) and IPS on four representative benchmarks.

| Method | Bamboogle | 2Wiki | HotpotQA | GAIA |
|---|---|---|---|---|
| w/ MSA | 56.27 | **52.33** | 50.67 | 11.55 |
| w/ IPS | **58.13** | 51.83 | **61.00** | **15.22** |

Results in Table 3 reveal a clear task-dependent pattern. On 2Wiki, a predominantly search-oriented benchmark with limited reasoning depth, MSA slightly outperforms IPS (52.33% vs. 51.83%). In this setting, performance is largely driven by information retrieval, and averaging feasibility estimates provides a reasonable approximation.

In contrast, IPS consistently outperforms MSA on tasks requiring multi-step reasoning and exploration. On HotpotQA and GAIA, IPS achieves gains of 10.33% and 3.67%, respectively. These tasks exhibit stronger agent-level disagreement, where naive score averaging masks inconsistencies across reasoning paths and leads to overconfident yet weakly grounded plan selection. By favoring plans that remain consistent across agent assessments, IPS enables EPC-AW to better mitigate miscalibration in complex planning regimes.

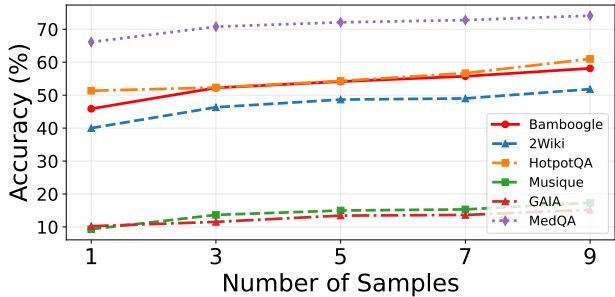

*Figure 2.* Sensitivity of EPC-AW to the number of sampled candidate plans.

### 5.5. Hyperparameter Sensitivity

We analyze the sensitivity of EPC-AW to the number of sampled plans $n$ in IPS, varying $n \in \{1, 3, 5, 7, 9\}$ across all datasets. When $n = 1$, IPS degenerates to generating a single plan under heterogeneous information, leaving no alternative plans under different knowledge states for comparison. As a result, EPC-AW cannot assess cross-agent consistency and yields the weakest performance.

Once $n > 1$, performance improves markedly, with the largest gain observed from $n = 1$ to $n = 3$, indicating that the main benefit comes from introducing epistemic diversity rather than exhaustive sampling.

As $n$ further increases, performance continues to improve monotonically with diminishing returns, as additional samples provide denser coverage of the epistemic space and enable more reliable epistemic calibration. This consistent trend also suggests that EPC-AW is robust to the choice of $n$ and does not rely on delicate hyperparameter tuning.

Overall, EPC-AW exhibits a stable and monotonic performance trend with respect to $n$, demonstrating that its gains stem from improved epistemic calibration rather than sensitive hyperparameter tuning.

*Table 4.* Generalization results across different LLM backbones.

| Method | Qwen3-14B | | | | DeepSeek-R1-32B | | | |
|---|---|---|---|---|---|---|---|---|
| | Bamboogle | 2Wiki | HotpotQA | GAIA | Bamboogle | 2Wiki | HotpotQA | GAIA |
| No-Repair | 45.60 | 36.50 | 42.00 | 6.29 | 48.80 | 43.00 | 49.00 | 7.87 |
| Retry | 48.00 | 38.00 | 49.00 | 7.87 | 51.20 | 44.50 | 53.00 | 9.44 |
| Rollback | 48.80 | 42.50 | 51.00 | 7.87 | 53.60 | 47.50 | 57.00 | 10.24 |
| EPC-AW | **56.80** | **48.50** | **58.00** | **11.81** | **62.40** | **52.00** | **63.00** | **15.75** |

### 5.6. Generalization Across LLM Backbones

To evaluate whether the effectiveness of EPC-AW depends on a specific backbone, we further conduct experiments on two additional LLMs: Qwen3-14B and DeepSeek-R1-32B. As shown in Table 4, EPC-AW consistently improves performance across all datasets and model architectures. Compared with No-Repair, EPC-AW yields substantial gains on both backbones, with average improvements of 11.18% on Qwen3-14B and 11.13% on DeepSeek-R1-32B.

These results suggest that the effectiveness of EPC-AW does not rely on backbone-specific behaviors. Instead, EPC-AW consistently improves the epistemic calibration process during multi-agent planning and execution, enabling agents to better identify unreliable plans, revise intermediate reasoning trajectories, and recover from latent planning failures. Overall, the results demonstrate that EPC-AW generalizes across heterogeneous LLM architectures.

### 5.7. Time and Token Cost Analysis

We analyze the computational overhead of EPC-AW from both time and token perspectives. In agentic workflows, the dominant time cost mainly comes from external tool execution and LLM inference, while the dominant token cost comes from accumulated memory and retrieved evidence.

**Base Workflow.** Suppose the system runs for $T$ interaction rounds. Let $\tau_{\text{llm}}$ denote the average latency of one LLM call and $\tau_{\text{tool}}$ denote the latency of external tool execution. Each interaction round consists of one planning call, one execution step, and diagnosis-related LLM feedback. The overall time complexity is therefore $\mathcal{O}\big(T \cdot (\tau_{\text{llm}} + \tau_{\text{tool}})\big)$.

For token complexity, let $m_i^{(t)}$ denote the memory token length of agent $i \in \{p, e, d\}$ at round $t$, where $p$, $e$, and $d$ correspond to the Planner, Executor, and Diagnoser, respectively. Let $q_i$ denote the corresponding prompt token length and $n_p$ denote the average plan length. The token complexity of the base workflow is

$$\mathcal{O}\left( \sum_{t=1}^{T} \Big( \sum_{i \in \{p,e,d\}} m_i^{(t)} + \sum_{i \in \{p,e,d\}} q_i + n_p \Big) \right). \quad (25)$$

**EPC-AW.** The additional cost in EPC-AW is from IPS and CESR. IPS introduces two additional components: (i) generation of $K$ candidate plans (single LLM call producing $\mathcal{O}(Kn_p)$ output tokens), and (ii) six evaluation calls (self- and cross-agent), each processing all $K$ plans. Since these evaluation calls can be executed in parallel, the asymptotic time complexity remains $\mathcal{O}\big(T \cdot (\tau_{\text{llm}} + \tau_{\text{tool}})\big)$. The token complexity becomes

$$\mathcal{O}\left( \sum_{t=1}^{T} \Big( \sum_{i \in \{p,e,d\}} m_i^{(t)} + \sum_{i \in \{p,e,d\}} q_i + Kn_p \Big) \right), \quad (26)$$

where the additional $Kn_p$ term reflects multi-plan reasoning and evaluation. CESR does not introduce additional LLM calls and only adds a constant number of constraint tokens to the Diagnoser input. Therefore, it does not change the asymptotic time or token complexity.

Overall, EPC-AW preserves asymptotic complexity while improving success rate (+9.75% avg.), offering a favorable cost–performance trade-off. Notably, the additional cost primarily scales the accumulated memory and retrieved context, which already dominate token consumption in the base system, rather than introducing a new computational bottleneck. Further details and statistics on tool usage and inference are provided in Appendix E.

## 6. Conclusion

We identify epistemic miscalibration during planning as a distinct and previously underexplored failure mode in LLM-based multi-agent systems. Unlike execution errors, such miscalibration remains latent and dynamic during planning, as plans can appear executable while feasibility assessments evolve with newly acquired information. By shifting mitigation from execution-time correction to planning-time calibration, our approach improves overall system reliability. EPC-AW achieves the best results on all benchmarks, yielding a 9.75% absolute improvement on the system-level accuracy. These results underscore the importance of treating epistemic calibration as a first-class, system-level consideration in the design of robust multi-agent workflows. Looking forward, epistemic calibration provides a promising perspective for improving LLM-based multi-agent systems.

## Acknowledgments

This work was supported in part by the National Natural Science Foundation of China under Grant 62572346. This work was also supported by the Joint Laboratory of AI for Smart Education, Gaotu-RUC, and the Shanghai Key Laboratory of Data Science.

## Impact Statement

This paper presents work whose goal is to improve the correctness and reliability of LLM-based multi-agent systems by mitigating epistemic miscalibration during planning. By reducing unsupported feasibility assessments, the proposed approach improves task success in settings with incomplete and evolving information, and may contribute to more reliable deployment of LLM-based systems in applications where planning quality directly affects outcomes. The work advances system-level planning reliability without introducing new capabilities or ethical concerns beyond those already well studied in large language model deployment.

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

# A. Dataset Details

We evaluate our approach on a diverse set of benchmarks that span a broad spectrum of dependence on internal inference versus external information retrieval.

**Bamboogle** (Press et al., 2023) is a manually constructed multi-step reasoning dataset designed to probe compositional inference over interconnected facts. Each question typically requires up to four explicit inferential steps.

**2Wiki** (Ho et al., 2020) combines structured knowledge from Wikidata with unstructured evidence from Wikipedia, requiring models to retrieve and integrate information across heterogeneous sources. While annotated reasoning chains facilitate interpretable multi-hop inference, successful reasoning is contingent on accurate intermediate retrieval, making the dataset jointly dependent on search quality and sequential reasoning.

**HotpotQA** (Yang et al., 2018) is a widely used multi-hop question answering benchmark constructed from Wikipedia articles. Questions require integrating information from multiple documents within a relatively homogeneous textual space.

**Musique** (Trivedi et al., 2022) is a challenging multi-step reasoning dataset characterized by strong sequential dependencies, where each inference stage relies critically on conclusions derived from prior steps.

**GAIA** (Mialon et al., 2023) is a benchmark designed to assess general AI agents in open-world settings, requiring capabilities such as multi-step reasoning, web navigation, and tool use. Unlike conventional QA datasets, GAIA emphasizes planning and information acquisition under partial observability, with reasoning tightly coupled to search and tool utilization.

**MedQA** (Yang et al., 2024) consists of multiple-choice questions derived from professional medical licensing examinations. The dataset evaluates clinical reasoning grounded in domain knowledge, requiring models to infer diagnoses or treatments through structured medical reasoning.

Overall, these datasets collectively cover a continuum from reasoning-dominant inference to search-driven open-world problem solving. This diversity allows us to systematically examine how epistemic miscalibration manifests across different planning and reasoning conditions in LLM-based multi-agent systems.

We follow AgentFlow (Li et al., 2026) and directly adopt its evaluation subsets for all datasets to ensure comparability with prior agent-based reasoning systems. Details are shown in Table. 5.

*Table 5.* Evaluation subsets adopted from AgentFlow. Each dataset occupies a single row, and all samples are uniformly drawn from the official splits and fixed across all methods.

| Dataset | Bamboogle | 2Wiki | HotpotQA | Musique | GAIA | MedQA |
|---|---|---|---|---|---|---|
| **#Samples** | 125 | 200 | 100 | 200 | 127 | 300 |

# B. Baseline Details

This section provides detailed descriptions of the baselines used in our experiments. Notably, neither MAST (Shen et al., 2025) nor COCO (Liang et al., 2025) has released official implementations at the time of writing. Accordingly, all baselines are our re-implementations inspired by the methodological descriptions in the original papers, instantiated within the AgentFlow framework (Li et al., 2026). All methods share the same agent architecture, system memory design, and execution protocol, ensuring a controlled and fair comparison.

Across all baselines, agents maintain access to a shared system memory that records the complete execution history, including generated plans (goals and actions) and corresponding execution results. For baselines involving epistemic assessment, diagnosis is performed exclusively during the planning phase.

**No-Repair (AgentFlow).** The No-Repair baseline directly follows the original AgentFlow design. The agent generates a complete plan in a single forward pass and executes it without any explicit feedback, diagnosis, or recovery mechanism. This baseline serves as a reference for unmitigated epistemic miscalibration.

**Retry (Planning-Level Metacognitive Re-Generation).** The Retry baseline adapts metacognitive retry strategies originally proposed in MAST (Shen et al., 2025) to the planning stage. At each planning step, the agent evaluates whether the

newly generated plan is epistemically miscalibrated based on the execution history up to the current step. If miscalibration is detected, the agent re-generates the current planning step while preserving all prior history. The diagnostic analysis is explicitly provided as feedback to guide the re-generation. Importantly, Retry does not modify the global system state or historical memory; it only replaces the current planning output.

**Rollback (System-Level State Recovery).** The Rollback baseline is inspired by the contextual rollback mechanism in COCO (Liang et al., 2025). When epistemic miscalibration is detected at the current planning step, the agent first determines an appropriate historical step to revert to. The entire system state, including shared memory and recorded execution history, is rolled back to the selected step. Planning is then re-executed from that point onward, with the diagnostic analysis provided as feedback to guide re-planning. Compared to Retry, Rollback performs a global state recovery rather than a local re-generation, allowing the system to correct errors that may have propagated across multiple planning steps.

Notably, Retry and Rollback differ fundamentally in their scope of intervention. Retry performs a local correction at the current planning step without altering historical state, while Rollback restores the system to a previous state and re-optimizes subsequent planning decisions. Both baselines operate under identical detection criteria and planning-phase constraints, enabling controlled evaluation of their effectiveness in mitigating epistemic miscalibration.

## C. Implementation Details

This section provides additional implementation details to support reproducibility and clarify key design choices.

**Backbone Model.** All agents in EPC-AW, including the Planner, Executor, and Diagnoser, are instantiated using Qwen3-Coder-30B (Yang et al., 2025). The model is deployed on a server equipped with four NVIDIA RTX 4090 GPUs and served via `vLLM` for efficient inference. This backbone is selected to accommodate the long-context requirements of multi-step agentic planning and execution, where intermediate plans, tool interactions, and execution traces must be preserved across extended reasoning horizons. To ensure a fair comparison, the same backbone model and deployment configuration are used consistently across EPC-AW and all baseline methods.

**Planning and Generation Settings.** During the planning phase, the Planner generates a fixed number of $n = 9$ candidate next-step plans. To encourage exploration of alternative planning hypotheses, candidate generation is performed with temperature $0.9$. All subsequent generations, including execution, diagnosis, and tool-based reasoning, are performed with temperature $0$ to ensure deterministic behavior and reduce variance across runs. The maximum number of iterations is set to 10 for all experiments.

In IPS, multi-agent plan evaluation and belief prediction are performed independently and in parallel. Candidate plans are evaluated using a predefined feasibility metric, where agents assign integer scores from 1 to 5 reflecting their epistemic assessment of plan feasibility.

---

**Feasibility Metric**

**Score 5 — Exceptional Feasibility**
The plan is internally coherent, precise, and strongly justified.

- Tool selection and parameters are fully specified and sufficient *in principle* to support the stated sub-goal under the given context.

- Reasoning is complete, logically tight, and optimally grounded in the available information.

- No implicit assumptions or missing steps are required to interpret the plan.

**Score 4 — Near-Perfect Feasibility**
The plan is coherent and well-aligned with the stated sub-goal.

- Tool selection is correct; parameters are appropriate but may allow minor refinement.

- Reasoning is sound, though some details could be made more explicit.

- The plan is interpretable without major inference.

---

**Score 3 — Strong Feasibility**
The plan is plausible and directly addresses the sub-goal.

- Tool selection is mostly correct; some parameters or steps require mild inference.

- Reasoning is generally sound but partially underspecified.

- The plan remains interpretable, though not maximally precise.

**Score 2 — Mostly Feasible**
The plan is relevant but exhibits notable epistemic gaps.

- Tool selection is reasonable, but parameters are under-specified or ambiguous.

- Reasoning relies on implicit assumptions or missing details.

- Additional clarification would be required for confident interpretation.

**Score 1 — Weak Feasibility**
The plan shows limited coherence or weak alignment with the sub-goal.

- Tool selection or parameter specification is incomplete or mismatched.

- Reasoning is vague, fragmented, or poorly grounded in the given context.

- The intended effect of the plan is epistemically unclear.

**Tool Configuration.** Following AgentFlow (Li et al., 2026), the system interacts with five tools: 1) a base generator for default reasoning, 2) a Python coder that generates and executes Python code and returns execution results, 3) Google Search for retrieving relevant web pages, 4) Wikipedia Search for structured knowledge retrieval, and 5) Web Search for page-level information extraction and summarization.

Compared to the original AgentFlow implementation, we replace direct Google Search API calls with a lightweight local retrieval pipeline that collects the top-10 relevant web pages via a local Chrome-based Google Search interface. This modification avoids potential interference from backend LLMs (e.g., Gemini-based models) implicitly involved in official API responses, ensuring that information retrieval does not benefit from stronger external language models.

To further mitigate information leakage due to publicly available datasets, we apply keyword-based filtering during the search stage to exclude HuggingFace repositories and dataset-specific pages. Retrieved pages are then processed by the Web Search tool, followed by LLM-based summarization.

**Evaluation Protocol.** For evaluation, GPT-4o is employed as an automatic judge to determine whether model predictions match the corresponding ground-truth answers, following standard practice in tool-augmented reasoning benchmarks (Li et al., 2026; Arif et al., 2024). To reduce the impact of stochasticity, all experiments are repeated three times, and we report the average accuracy across runs. All evaluation settings are aligned with those used in AgentFlow to facilitate direct comparison.

## D. EPC-AW Pseudocode and Explanation

The EPC-AW algorithm iteratively performs planning, execution, and diagnosis in a multi-agent workflow to improve epistemic calibration. The workflow can be described as follows:

- **Initialization:** System memory $\mathcal{M}_{\text{sys}}$ stores the query $Q$, execution history, and the abstract description of the role for all agents $\mathcal{U}$. Each agent (Planner $P$, Executor $E$, Diagnoser $D$) maintains a private memory $\mathcal{M}_i$. The evidence set $\Psi$ is initialized empty.
- **Candidate Plan Generation:** The Planner generates a set of candidate plans $\Pi$ based on the system memory and its policy $\gamma_P$.
- **Plan Selection:** IPS chooses the most epistemically promising plan $\pi^*$ from $\Pi$, taking into account cross-agent

---

**Algorithm 1** EPC-AW: Epistemic Planning Calibration Agentic Workflow

---

    **Input:** query $Q$, maximum rounds $T$
    **Output:** final answer $Y$
    Initialize system memory $\mathcal{M}_{\text{sys}} \leftarrow \langle Q, \emptyset, \mathcal{U} \rangle$
    Initialize agent memories $\mathcal{M}_P \leftarrow \emptyset, \mathcal{M}_E \leftarrow \emptyset, \mathcal{M}_D \leftarrow \emptyset$
    Initialize evidence set $\Psi \leftarrow \emptyset$
    **for** $t = 1$ **to** $T$ **do**
        Generate candidate plans $\Pi \leftarrow A_P(\mathcal{M}_{\text{sys}}, \gamma_P)$
        Select plan $\pi^* \leftarrow \text{IPS}(\Pi, \mathcal{M}_{\text{sys}}, \{\gamma_i\})$
        Execute plan and observe outcome $o \leftarrow A_E(\pi^*)$
        Diagnose execution $d \leftarrow A_D(\pi^*, o)$
        **if** $\mathbf{1}_{\text{fail}}(d) = 0$ **then**
            Extract evidence $\psi \leftarrow \mathcal{F}(d)$
            $\Psi \leftarrow \Psi \cup \{\psi\}$
        **end if**
        Compute planner-selected plan $\pi_P \leftarrow \arg\max_{\pi_k \in \Pi} e_P(k)$
        Update planner memory $\mathcal{M}_P \leftarrow \text{CESR}(\pi_P, \pi^*, \mathcal{M}_{\text{sys}}, \mathcal{M}_P)$
        **if** $\mathbf{1}_{\text{stop}}(Q, \Psi) = 1$ **then**
            **return** $\mathcal{G}(Q, \Psi, \mathcal{H})$
        **end if**
    **end for**
    **return** $\mathcal{G}(Q, \Psi, \mathcal{H})$

---

alignment.

- **Plan Execution and Diagnosis:** The Executor executes $\pi^*$ and observes outcome $o$. The Diagnoser evaluates the execution outcome $o$ whether it satisfies the plan $\pi^*$.
- **Evidence Extraction:** If the execution satisfies the plan $\pi^*$ ($\mathbf{1}_{\text{fail}}(d) = 0$), evidence $\psi$ is extracted and added to $\Psi$.
- **Planner Memory Update:** The Planner identifies its selected plan $\pi_P$ and updates its memory $\mathcal{M}_P$ via CESR, aligning plan selection with accumulated feedback.
- **Termination Condition:** The algorithm stops if the accumulated evidence $\Psi$ suffices to answer query $Q$ ($\mathbf{1}_{\text{stop}}(Q, \Psi) = 1$), returning the final answer $\mathcal{G}(Q, \Psi, \mathcal{H})$.
- **Iteration:** If termination criteria are not met, the loop continues until maximum rounds $T$ are reached.

This workflow ensures that the multi-agent system iteratively refines plans and updates epistemic states, leading to more reliable final answers under uncertainty.

## E. Time and Token Cost Analysis

In this section, we provide a detailed analysis of the computational overhead introduced by EPC-AW. From the system perspective, the overall time cost is mainly dominated by external tool execution and LLM inference latency, while the token cost is primarily determined by accumulated memory, retrieved evidence, and long-context reasoning. We therefore separately analyze how IPS and CESR affect these two aspects.

### E.1. Base Workflow Complexity

Suppose the system runs for $T$ interaction rounds. At each round, the Planner generates a plan, the Executor invokes external tools, and the Diagnoser performs failure analysis and feedback generation. Let $m_i^{(t)}$ denote the memory token length of agent $i \in \{p, e, d\}$ at round $t$, where $p$, $e$, and $d$ correspond to the Planner, Executor, and Diagnoser, respectively. Let $q_i$ denote the fixed prompt token length associated with each agent, and let $n_p$ denote the average token length of a generated plan. We further denote by $\tau_{\text{llm}}$ the average latency of a single LLM call and by $\tau_{\text{tool}}$ the latency of external tool execution.

Under this formulation, the dominant runtime cost at each interaction round comes from LLM inference and external tool usage. Therefore, the overall time complexity can be written as $\mathcal{O}\big(T(\tau_{\text{llm}} + \tau_{\text{tool}})\big)$. The corresponding token complexity is

$$\mathcal{O}\left( \sum_{t=1}^{T} \left( \sum_{i\in\{p,e,d\}} m_i^{(t)} + \sum_{i\in\{p,e,d\}} q_i + n_p \right) \right), \tag{27}$$

where the first term corresponds to accumulated memory and retrieved evidence, the second term represents fixed prompt instructions, and the final term corresponds to the generated planning content. In practice, the dominant token source mainly comes from growing memory and retrieved external evidence rather than the plan itself.

### E.2. EPC-AW Complexity

**IPS Overhead.** IPS introduces two additional operations: multi-plan generation and epistemic evaluation. Specifically, the Planner generates $K$ candidate plans, producing approximately $\mathcal{O}(Kn_p)$ output tokens, followed by six self-/cross-agent evaluation calls that jointly assess feasibility consistency across agents.

Although IPS increases the number of LLM calls, these evaluation calls are mutually independent and can therefore be executed fully in parallel. As a result, the asymptotic time complexity remains $\mathcal{O}\big(T(\tau_{\text{llm}} + \tau_{\text{tool}})\big)$, with only a constant-factor increase in practical latency.

The token complexity becomes

$$\mathcal{O}\left( \sum_{t=1}^{T} \left( \sum_{i\in\{p,e,d\}} m_i^{(t)} + \sum_{i\in\{p,e,d\}} q_i + Kn_p \right) \right). \tag{28}$$

Compared with the base workflow, the only additional term is the multiplicative factor $K$ applied to the planning output. Since $K$ is treated as a small constant in practice, IPS does not change the asymptotic scaling behavior of the system.

Importantly, IPS mainly amplifies an already dominant token source rather than introducing a fundamentally new cost component. In realistic multi-hop workflows, retrieved evidence already contributes substantial context length (approximately 800–1000 tokens per step), while accumulated memory continuously grows across interaction rounds. Therefore, the additional overhead introduced by IPS primarily scales existing context processing.

**CESR Overhead.** Compared with IPS, CESR introduces negligible computational overhead. CESR does not require additional planning branches or extra LLM calls, and only appends a constant number of consistency-related constraint tokens to the Diagnoser input. Therefore, CESR does not change either the asymptotic time complexity or the asymptotic token complexity of the workflow.

### E.3. System-Level Runtime Analysis

From a practical systems perspective, overall latency is mostly dominated by external tool usage. Table 6 summarizes the average latency of different workflow components in our implementation.

Among all operations, Google Search incurs the highest latency, requiring approximately $80s$ on average due to multi-hop webpage retrieval and downstream browsing. Wikipedia retrieval requires approximately $40s$ depending on the number of retrieved pages. In contrast, a single 30B-scale LLM inference call only requires approximately $3s$.

Therefore, although IPS introduces additional evaluation calls, its contribution to end-to-end latency remains relatively small, especially under parallel execution. In practice, IPS mainly introduces a constant-factor increase in token usage (approximately $2\times$) due to multi-plan generation and evaluation, while preserving the original asymptotic scaling behavior of the system.

Overall, EPC-AW preserves asymptotic complexity while improving success rate (+9.75% avg.), offering a favorable cost–performance trade-off. Notably, the additional cost primarily scales the accumulated memory and retrieved context, which already dominate token consumption in the base system, rather than introducing a new computational bottleneck.

*Table 6.* Time Cost Breakdown of EPC-AW Components

| Component / Tool | Average Latency (s) |
|---|---|
| Base LLM generator | 3 |
| Google Search | 80 |
| Wikipedia Search | 40 |
| Python Coder / Web Search | 5 |
| LLM Feedback (planning / execution / diagnosis) | 3 |
| Retry / Rollback (within one round) | 3 |
| EPC-AW (within one round) | Slightly above 3 (parallelizable) |

# F. CESR Case Studies

To further illustrate the lightweight epistemic constraints generated in CESR, we provide two representative examples from real execution traces.

**Case 1: Tool capability mismatch.** In `bamboogle 8.json`, the Planner assumes that a specific ScienceDirect URL is accessible and can provide primary scientific evidence. However, the Executor returns no usable content, revealing a mismatch between the assumed and actual capability of the retrieval tool. CESR diagnoses this failure as tool information mismatch and redirects planning toward feasible and verifiable alternatives.

> **Generated Constraint**
>
> When a specific tool or URL fails to retrieve required evidence, do not assume alternative paths are equally viable without validation; instead, revise the plan toward achievable and verifiable sources.

**Case 2: Infeasible verification under information limits.** In `bamboogle 11.json`, the Planner repeatedly issues Google searches assuming that a complete list of Theranos whistleblowers and their connections to senior U.S. government officials can be exhaustively verified. However, the Executor results show that such authoritative and comprehensive evidence is unavailable, exposing a mismatch between assumed and actual verifiability. CESR identifies this as an unsupported sufficiency assumption and revises the objective to an epistemically reachable scope, focusing instead on confirmed whistleblowers (e.g., Tyler Shultz and Erika Cheung) using reliable sources such as Wikipedia. This revision enables successful task completion.

> **Generated Constraint**
>
> When a task requires verifying relationships or affiliations dependent on authoritative or inaccessible sources, avoid assuming such information is fully obtainable; instead, revise the goal to align with epistemically reachable and verifiable evidence.

# G. Related Work

### G.1. Failure and Repair in LLM-based Multi-agent Systems

LLM-based multi-agent systems have been widely adopted for complex decision making, tool use, and long-horizon task execution (Zhou et al., 2026; Ronanki, 2025; Lin et al., 2025). Despite their capabilities, these systems remain prone to failures arising from reasoning errors, coordination breakdowns, and unreliable or inconsistent use of information (Cemri et al., 2025; Hammond et al., 2025; Zhang et al., 2025a). Such failures can propagate across agents and rounds, often compounding errors in long-horizon tasks and multi-step planning scenarios. As a result, a growing body of work (Wan et al., 2025; Gu et al., 2025; Gong et al., 2025) has studied how to diagnose, localize, and repair failures in these systems. Most existing approaches focus on improving reliability by analyzing execution traces, monitoring agent interactions, or applying interventions at different stages of system runtime (Liang et al., 2025; Shen et al., 2025; Epperson et al., 2025; Ma et al., 2025), yet challenges remain in achieving robust cross-agent consistency and epistemic calibration, particularly when

agents hold heterogeneous or partial information.

Several methods have been proposed to address failures in LLM-based multi-agent systems, broadly falling into post-hoc or cross-run debugging, and online correction during execution. In the post-hoc category, AGDebugger (Epperson et al., 2025) allows developers to inspect and edit multi-agent interaction traces after execution, enabling identification and manual correction of failures. DoVer (Ma et al., 2025) further formalizes this approach by testing failure hypotheses through targeted interventions and validating their effects across repeated executions, effectively providing a controlled mechanism for debugging and improving system robustness over multiple runs.

In contrast, online correction methods aim to detect and mitigate failures as workflows unfold. COCO (Liang et al., 2025) applies continuous monitoring, rollback, and reflective reasoning to correct execution drift and resolve inter-agent inconsistencies in real time. Similarly, MASC (Shen et al., 2025) leverages history-conditioned anomaly signals to detect problematic execution steps and applies local corrections to prevent cascading errors that could compromise long-horizon tasks. While these methods improve reliability during runtime, they typically operate on observed traces or local signals without explicitly modeling cross-agent epistemic miscalibration, leaving challenges in ensuring robust coordination and consistent belief updating across heterogeneous agents.

Despite differences in how and when repair is applied, these methods mainly target incorrect actions or local reasoning errors but neglect the plan's commitments, allowing unreliable feasibility assessments to persist. In contrast, this work formulates *epistemic miscalibration in planning* as a distinct repair target in LLM-based multi-agent systems. It reveals that system-level failures can arise from miscalibrated feasibility assessments at planning time, even when all subsequent executions are locally correct.

### G.2. Model- and Agent-Level Epistemic Calibration

A growing body of work studies epistemic calibration in large language models (LLMs), focusing on overconfidence (Wen et al., 2024; Hagar et al., 2025; Tripathi et al., 2025) and uncertainty estimation (Liu et al., 2025; Wang et al., 2025; Lee et al., 2025; Nguyen-Hien et al., 2025). This line of research primarily treats epistemic miscalibration as a property of an individual model or agent, aiming to align a model's expressed confidence with its actual knowledge. Methods include post-hoc confidence adjustment, uncertainty-aware decoding, and calibration-aware fine-tuning, which have been shown to reduce overconfident errors and improve the reliability of single-agent predictions. Even when multiple agents are considered, the focus largely remains on the calibration of individual LLM judgments rather than on the interactions or coordination of a deployed multi-agent system.

Several approaches leverage multi-agent structures as a means to improve agent-level calibration (Clark et al., 2025; Wen et al., 2024). Consensus or voting mechanisms aggregate agents' assessments to stabilize judgments (Pitre et al., 2025; Wen et al., 2026), while verifier agents critique intermediate outputs based on detectable errors or invalid responses (Sung et al., 2025; Gupta, 2025). These methods rely on LLMs acting as judges to evaluate content quality, but such first-order judgments are themselves subject to epistemic miscalibration, which can propagate errors and limit their ability to reliably diagnose failures. Additionally, aggregation-based approaches may smooth out but not eliminate systemic biases shared across agents, leaving residual miscalibration unaddressed.

Another related line of work draws from peer prediction and game-theoretic truth inference (Cho et al., 2025; Kim et al., 2025). Bayesian Truth Serum (Witkowski & Parkes, 2012; Weaver & Prelec, 2013) and its extensions elicit truthful reporting by rewarding agreement with peers' predictions. Recent adaptations implement explicit scoring rules or discrimination games for LLM-based agents (Chen et al., 2025), aiming to incentivize accurate and self-consistent outputs. While effective under static Bayesian game assumptions, these approaches rely on repeated interactions, payoff signals, or structured incentives, and thus do not naturally extend to dynamic multi-agent systems with heterogeneous knowledge, limited interventions, and long-horizon task dependencies.

In contrast, we study epistemic miscalibration as a source of failure in LLM-based multi-agent systems. Our goal is to repair task failures by intervening during system operation, where there is no additional payoff signals or external supervision. Inspired by peer-based calibration methods, we instead exploit the stability of agents' evaluations across heterogeneous information and refine epistemic states over time based on persistent cross-agent inconsistencies.

