# OpenReview forum: "When Planning Fails Despite Correct Execution: On Epistemic Calibration for LLM-Based Multi-Agent Systems"
_ICML.cc/2026/Conference — ICML 2026 regular_

### Official Review · Reviewer_BSFA · 2026-03-01

**Soundness:** 3
**Presentation:** 3
**Significance:** 3
**Originality:** 3
**Overall Recommendation:** 5
**Confidence:** 3

**Summary:**

The paper explores the problem when an LLM agents generates a plan that fails when it is executed because of a  "epistemic miscalibration" when the plan was generated. The paper, as pointed by the authors, is in a research direction for multi-agent systems that concern execution failures. There is a need to detect and correct such failures. Frequently, such systems monitor errors seeking to intervene before acatastrophic failure occurs. This paper is exploring failures that are not execution failures. For instance, despite all actions being executed correctly, the input task remains unachieved. This may be due to gaps in the system's knowledge that the agent is failing to detect. This can be exacerbated un dynamic environments where new information is acquired. The key idea is, instead of plan verification, it checks if the plan is still supported.

**Compliance With Llm Reviewing Policy:**

Affirmed.

**Key Questions For Authors:**

See the issue abiove regarding J(...) and \tilde{Y}

**Limitations:**

No limitations are discussed but I don't see an issue here.

**Strengths And Weaknesses:**

The idea of evaluating, in Information-consistency-based Plan Selection (IPS), if the plan is feasible across agents is sound. The system detects if the plan evaluation varies significantly across agents, in the Consistency-guided Epistemic State Refinement (CESR).

The related work section is well done, discussing failures and repairs in the context of LLM-based planning, and epistemic calibration in LLMs. The distinction of the work in the paper versus the former is the focus on epistemic miscalibration instead of failures. I couldn't quite understand the distinction versus the latter. I agree that LLMs themselves may result in miscalibration but can't parse: "we study epistemic miscalibration as a system-level failure ". Not sure what system-level means here.

The formalization is a function "D(.)" of the annotated plan with goals, the previously generated information context and the evidence.  The subject feasibility J(plan∣information) suggest a probabilistic/belief framework. But the paper doesn't introduce what the probabilistic/belief model is. At the same time, correctness is defined deterministically: if \tilde{Y} = Y*. I am unclear if J(plan∣information)  is a probability or a belief distribution (if so which?) .

The figure helps understand the flow between the planner, the executor and the diagnoser. The system maintains the full  interaction history H including the outcomes and the diagnostic of the outcomes. It also has a memory M of the query, the verifiable evidense and role descriptions; M is  a projection of the history dependent on the roles. The planner generates candidate plans based on the system memory. I grasp what is been done here but can't fully evaluate the details.

The evaluation uses a variety of standard  reasoning and search benchmarks typical of LLM mutil-agent systems. The types of baselines are also standard in LLM-based MAS. The result suggest strong performance by the system compared to its baselines. The ablation results are also strong.

---

> ### Author Rebuttal · Authors · 2026-03-30
>
> We thank the reviewer for the positive assessment of IPS/CESR and for raising these important questions regarding the conceptual distinction and formalization.
>
> ---
>
> ### **1. Clarifying system-level failure**
>
> We clarify that the notion of **system-level failure** refers to the final task failure ($\hat{Y} \neq Y^*$).
> In the related work, we point out that existing approaches mainly focus on repairing execution-stage errors to prevent system-level failures. In contrast, **we study epistemic miscalibration as a system-level failure mode that arises during the operation of LLM-based multi-agent systems.**
>
> Our focus is on epistemic miscalibration at the planning stage leads to infeasible plans, which, despite correct execution of all actions, result in failure to achieve the task objective.
>
> ---
>
> ### **2. Clarifying the role of $J(\pi \mid \mathcal{I})$**
>
> We appreciate the reviewer for pointing out this ambiguity. $J(\pi \mid \mathcal{I})$ is **not** intended to represent a formal probabilistic or Bayesian belief model. Instead, it denotes an **implicit feasibility assessment** produced by agents conditioned on their current information context.
> This assessment does not assume probabilistic semantics or a belief distribution, does not require statistical calibration, and is only required to be comparable across agents and information conditions.
>
> Our framework operates on these relative assessments (e.g., consistency across agents), rather than on absolute probability values. We have revised the paper to explicitly state that $J$ is an abstract scoring/judgment function to avoid confusion with probabilistic formulations.
>
> ---
> ### **3. Clarifying Workflow Details**
>
> We thank the reviewer for carefully reading and summarizing the system workflow. To address the concern regarding the evaluation of details, we have added more explanations and elaborated on several components of the workflow in the revision, providing additional information to help readers better understand the system’s operations.
>
> Furthermore, in Appendix D, we provide the full pseudocode of EPC-AW along with detailed descriptions of the whole workflow. These materials are intended to assist readers in fully comprehending the workflow and understanding the underlying algorithmic mechanisms, complementing the high-level diagram in the main text.

---

> > ### Author Rebuttal · Reviewer_BSFA · 2026-04-03
> >
> > I don't have any more clarification needs. My original assessment stands.

---

> > > ### Author Response · Authors · 2026-04-07
> > >
> > > We thank you for your feedback and for your careful evaluation of our responses.
> > > We appreciate your time and consideration in evaluating our work.

---

### Official Review · Reviewer_9LxC · 2026-03-10

**Soundness:** 3
**Presentation:** 3
**Significance:** 3
**Originality:** 4
**Overall Recommendation:** 4
**Confidence:** 2

**Summary:**

The paper defines the notion of Epistemic (mis)calibration, as used in planning, to model the situation where an agent's confidence about a plan being able to achieve a goal is not correct. As a second step, they define an agentic workflow aiming at assessing the veracity of its assumptions about the capability of a plan to achieve a goal. This is done by validating the plan under varying conditions.
The experiments show that the proposed technique does improve the system-level success.

**Compliance With Llm Reviewing Policy:**

Affirmed.

**Key Questions For Authors:**

Based on the considerations above, can you try to qualify/ quantify the extra effort needed to run your proposed approach, in relation to the gains in performance? This would make your approach much more convincing. It would be nice to have some explanation which goes beyond a simple quantitative measure of time. For instance relating the complexity of your approach to the complexity of the overall system

**Limitations:**

See the question to the authors

**Strengths And Weaknesses:**

-Strengths
- The idea is interesting and novel
- The proposed solution is well articulated and presented
- The experiments show some level of success, even though not a very high one (an average of 9.75% is claimed by the authors

Weakenesses
- The real applicability of the proposed solution is not clear. The process itself of epistemic calibration can fail for exactly the same reasons which motivate its use. A proper and successful use of the proposed method crucially depends on the agent having a good estimate of its level of (non)-understanding of the problem. But building this understanding requires time and resources. Of course, more information increases the probability of building a good plan. But what is the cost of generating such a plan? In certain cases, just running a plan and seeing what happens is faster than reasoning about how to do it (of course, in those situations where a failing plan does not cause further problems hard to solve). Your analysis of the time cost (Section 5.6) is part of the answer, but it is a little too brutal and simplistic to be fully convincing.

---

> ### Author Rebuttal · Authors · 2026-03-30
>
> We thank the reviewer for the insightful comments on applicability and cost-effectiveness.
>
> ---
>
> ### **1. On when calibration is beneficial.**
> We agree that in some scenarios, directly executing a plan may be more efficient than reasoning. However, **trial-and-error strategies often struggle to reliably recover, especially in complex multi-step settings, leading to reduced task success rates**.
>
> To clarify applicability, we further analyze performance across tasks of varying difficulty (Sec. 5.4). we compare IPS with a consensus-based baseline. Results reveal a clear task-dependent pattern: **IPS provides substantially larger gains on complex reasoning tasks (e.g., +10.33% on HotpotQA and +3.67% on GAIA)**, where incorrect plans tend to propagate and cause cascading failures.
>
> This suggests that EPC-AW is particularly beneficial in high-complexity or high-risk scenarios, where the cost of executing an incorrect plan dominates the additional reasoning overhead.
>
> ---
>
> ### **2. On the cost of epistemic calibration and system-level complexity.**
> We thank the reviewer for the concern on cost and pointing out that our original analysis (Sec. 5.6) was coarse-grained. In the revision, we substantially extend this section with a more fine-grained decomposition of both time and token complexity, explicitly comparing the base workflow and the additional overhead introduced by IPS and CESR.
>
> **1) Complexity Analysis**
>
> **a. Base workflow (without IPS/CESR).**
> Each round consists of one planning call, one tool execution, and two diagnosis-related LLM calls. At round $t$, let $m_i^{(t)}$ denote the memory token length of agent $i \in \\{p,e,d\\}$, and $q_i$ denote the corresponding prompt token length. Let $n_p$ be the average token length of a plan. We denote by $\tau_{\text{llm}}$ the average latency of a single LLM call, and by $\tau_{\text{tool}}$ the average latency of tool execution.
>
> The time complexity is
> $\mathcal{O}\big(T \cdot (\tau_{\text{llm}} + \tau_{\text{tool}})\big)$, and the token complexity is $\mathcal{O}\Bigg(\sum_{t=1}^{T} \Big(\sum_{i \in \\{p,e,d\\}} m_i^{(t)} + \sum_{i \in \\{p,e,d\\}} q_i + n_p \Big)\Bigg)$.
>
> **b. IPS overhead.**
> IPS introduces two additional components: (i) generation of $K$ candidate plans (single LLM call producing $\mathcal{O}(K n_p)$ output tokens), and (ii) six evaluation calls (self- and cross-agent), each processing all $K$ plans.
> Since these evaluation calls can be executed in parallel, the asymptotic time complexity remains $\mathcal{O}\big(T \cdot (\tau_{\text{llm}} + \tau_{\text{tool}})\big)$, i.e., only a constant-factor increase in practice. The token complexity becomes $\mathcal{O}\Bigg(\sum_{t=1}^{T} \Big(\sum_{i \in \\{p,e,d\\}} m_i^{(t)} + \sum_{i \in \\{p,e,d\\}} q_i + K n_p \Big)\Bigg)$, where the additional $K n_p$ term reflects multi-plan reasoning and evaluation.
>
> **c. CESR overhead.**
> CESR does not introduce additional LLM calls and only adds a constant number of constraint tokens to the Diagnoser input. Therefore, it does not change the asymptotic time or token complexity.
>
> **d. System-level comparison.**
> Comparing before and after introducing IPS/CESR:
>
> - **Time complexity:** unchanged at $\mathcal{O}(T(\tau_{\text{llm}}+\tau_{\text{tool}}))$ under parallel execution.
> - **Token complexity:** remains $\mathcal{O}(\cdot)$ in the asymptotic sense, as $K$ is a constant.
>
> In practice, IPS introduces a **constant-factor increase** in token usage due to multi-plan generation and additional evaluation calls, but does not change the overall scaling behavior of the system.
>
> **2) Cost Analysis**
>
> From a systems perspective, the overall wall-clock latency is **dominated by external tool usage**, rather than LLM inference. In realistic multi-hop pipelines (e.g., search → browse → summarize), a single Google Search call takes ~80s and Wikipedia retrieval ~40s, while each LLM call requires only ~3s (30B-scale model). Since IPS only adds parallelizable evaluation calls, its contribution to end-to-end latency is marginal.
>
> IPS does increase the number of LLM calls per step, resulting in a constant-factor increase in token usage (≈2×) due to multi-plan generation and evaluation. However, this overhead primarily scales existing context processing rather than introducing a new cost source. In practice, retrieved evidence already contributes substantial token volume (≈800–1000 tokens per step), and growing memory dominates input size across all calls. IPS thus amplifies this dominant component without fundamentally altering the system’s cost structure.
>
> Importantly, while IPS incurs a moderate additional computational cost, it significantly improves decision quality (+9.75% on average). **This leads to a favorable cost–performance trade-off, particularly in complex tasks where incorrect plans are costly and difficult to recover from.**

---

> > ### Author Rebuttal · Reviewer_9LxC · 2026-04-06
> >
> > they answered my questions to the point

---

> > > ### Author Response · Authors · 2026-04-07
> > >
> > > We are glad that our responses addressed your questions clearly, and we appreciate your careful review and consideration.

---

### Official Review · Reviewer_YHB2 · 2026-03-13

**Soundness:** 3
**Presentation:** 3
**Significance:** 4
**Originality:** 4
**Overall Recommendation:** 5
**Confidence:** 4

**Summary:**

## Objective
This paper addresses an underexplored failure mode in LLM-based multi-agent systems: planning errors that occur even when all actions are correctly executed.

## Key Concept
The authors call this phenomenon "epistemic miscalibration": the planning agent overestimates its confidence in the feasibility of a plan, without this error being visible at execution time. The problem is doubly difficult to detect because it generates no observable error signal, and evolves over time as the system acquires new information.

## Proposed Method
To address this, they propose EPC-AW, a workflow built around two complementary mechanisms:

* IPS (Information-consistency-based Plan Selection): selects plans whose evaluation remains stable across multiple agents operating under different information, signaling stronger epistemic grounding.
* CESR (Consistency-guided Epistemic State Refinement): records past divergences between agents to constrain future planning and prevent the same miscalibration patterns from recurring.


## Results
Tested on six reasoning benchmarks, the approach improves task success by an average of +9.75% compared to a system with no repair mechanism.

**Compliance With Llm Reviewing Policy:**

Affirmed.

**Final Justification:**

The rebuttal addressed my main concerns.

**Key Questions For Authors:**

Q1. Could the authors provide concrete examples of failures that occur without EPC-AW and that remain unresolved even with EPC-AW?
Q2. Have the authors tested EPC-AW against other types of LLMs?
Q3. Given that MAST and COCO baselines are re-implementations rather than official ones, how confident are the authors that these approximations faithfully reflect the original methods' performance?

**Limitations:**

Yes.

**Strengths And Weaknesses:**

## Strengths
* The paper is very clear and well-written.
* The problem is compellingly introduced and precisely defined, and the originality of both the challenge and the proposed approach is clearly established.
* The decomposition into two components allows for extensibility of the system while maintaining interpretability of EPC-AW's overall behavior.
* The experimental setup is solid and comprehensive.
* The technical analysis and ablation study are honest and rigorous, bringing strong credibility to the proposed approach. Results are reproducible.

## Weaknesses
* A more complete computational cost analysis seems necessary: the "Cost of Retry, Rollback and EPC-AW" section states that calls can be parallelized due to their independence, but this still requires specific infrastructure that may not be universally available — a concern that is particularly worth discussing given the use of an already non-trivial 30B model.
* On that note, it would be very interesting to conduct this study with other LLMs in order to compare results and better understand the limitations depending on the type of model used (reasoning-focused models, smaller/larger models, etc.).

---

> ### Author Rebuttal · Authors · 2026-03-30
>
> We sincerely thank the reviewer for the insightful comments and constructive suggestions.
>
> ---
>
> ### **1. Computational Cost Analysis and Practical Deployment (W1)**
> We appreciate the request for a more complete cost analysis and practical deployment.
> We clarify that the additional evaluation calls in IPS can be parallelized using modern LLM serving frameworks (e.g., vLLM) via batched inference within a single model instance. In such settings, multiple evaluation calls are executed concurrently with minimal additional latency.
>
> Importantly, even without parallel infrastructure, these additional evaluation calls introduce only a constant-factor increase in LLM latency, and thus do not change the overall time complexity of the system. In practice, however, **the overall wall-clock time is dominated by external tool calls rather than LLM inference**. Measurements show a single Google Search takes ~80 seconds and Wikipedia ~40 seconds, whereas each LLM call takes ~3 seconds on a 30B model.
>
> We have expanded both time and token complexity discussions.
>
> **1) Base workflow (without IPS/CESR).**
>
> Each round consists of one planning call, one tool execution, and two diagnosis-related LLM calls. At round $t$, let $m_i^{(t)}$ denote the memory token length of agent $I\in\\{p,e,d\\}$, and $q_i$ denote the corresponding prompt token length. Let $n_p$ be the average token length of a plan. Denote $\tau_{\text{llm}}$ and $\tau_{\text{tool}}$ as average LLM and tool latency.
>
> The time complexity is $\mathcal{O}(T\cdot(\tau_{\text{llm}}+\tau_{\text{tool}}))$, and the token complexity is
> $\mathcal{O}\Bigg(\sum_{t=1}^{T}\Big(\sum_{I\in\\{p,e,d\\}}m_i^{(t)}+\sum_{I\in\\{p,e,d\\}}q_i+n_p\Big)\Bigg)$.
>
> **2) IPS overhead.**
> IPS introduces two additional components: (i) generation of $K$ candidate plans (single LLM call producing $\mathcal{O}(Kn_p)$ output tokens), and (ii) six evaluation calls (self- and cross-agent), each processing all $K$ plans. Since these evaluation calls can be executed in parallel, the asymptotic time complexity remains $\mathcal{O}(T\cdot(\tau_{\text{llm}}+\tau_{\text{tool}}))$, i.e., only a constant-factor increase in practice. The token complexity becomes
> $\mathcal{O}\Bigg(\sum_{t=1}^{T}\Big(\sum_{I\in\\{p,e,d\\}}m_i^{(t)}+\sum_{I\in\\{p,e,d\\}}q_i+Kn_p\Big)\Bigg)$, where the additional $Kn_p$ term reflects multi-plan reasoning and evaluation.
>
> **3) CESR overhead.**
> CESR does not introduce additional LLM calls and only adds a constant number of constraint tokens to the Diagnoser input. Therefore, it does not change the asymptotic time or token complexity.
>
> Overall, **EPC-AW preserves asymptotic complexity while improving success rate (+9.75% avg.)**, achieving a favorable cost–performance trade-off. The added cost mainly scales existing memory and retrieved context, which are already the dominant token components, and does not introduce a new bottleneck.
>
> ---
>
> ### **2. Generalization Across LLMs (W2/Q2)**
>
> We thank the reviewer for the suggestion. We extended experiments to Qwen3-14B (smaller) and DeepSeek-R1-32B (reasoning-oriented) on two datasets. Results show consistent improvements, demonstrating that EPC-AW is model-agnostic and robust across both smaller and reasoning-enhanced LLMs.
>
> | Method | Qwen-14B (Bamboogle) | Qwen-14B (HotpotQA) | DeepSeek (Bamboogle) | DeepSeek (HotpotQA) |
> |---|----|---|----|---|
> | Non | 45.60| 42.00 | 48.80| 49.00 |
> | Retry | 48.00| 49.00 | 51.20| 53.00 |
> | Rollback | 48.80| 51.00 | 53.60| 57.00 |
> | EPC-AW | 56.80| 58.00 | 62.40| 63.00 |
>
> ---
>
> ### **3. Examples of Failures (Q1)**
>
> We appreciate this insightful question and provide representative failures:
> - **Tool-specific query loops** (id=11 in Bamboogle).
> When a retrieval returns weakly relevant results, the agent repeatedly issues near-identical queries using the same tool with slight parameter tweaks (e.g., “Theranos whistleblowers list”→“full list”→“complete list”), resulting in repeated calls without meaningful information gain.
> - **Parameter-driven multi-tool loops** (id=84 in Bamboogle).
> For plans that fail to retrieve useful information (e.g., searching for “Sigmund Freud”), the agent may cycle through different tools or tool settings (e.g., Wikipedia→Google Search) with minor query variations, failing to appropriately reflect on and adjust its search strategy based on the results obtained.
>
> We have included additional case studies to illustrate these behaviors in the revision.
>
> ---
>
> ### **4. Reliability of Re-implemented Baselines (Q3)**
>
> We acknowledge concerns about re-implementation fidelity. Since MAST and COCO lack official code, exact reproduction is infeasible; however, we faithfully implement their core mechanisms (retry for MAST, rollback for COCO) and adapt them to our setting with minimal optimizations (e.g., prompt tuning) that preserve strategies. All methods are evaluated within a unified framework, so performance differences mainly reflect the methods rather than implementation artifacts.

---

> > ### Author Rebuttal · Reviewer_YHB2 · 2026-04-02
> >
> > Thank you for the responses, it fully resolved my concerns.

---

> > > ### Author Response · Authors · 2026-04-07
> > >
> > > We sincerely thank you for your positive feedback and for acknowledging that our rebuttal has fully addressed the concerns.
> > > We greatly appreciate your time and thoughtful evaluation of our work.

---

### Official Review · Reviewer_tTHj · 2026-03-13

**Soundness:** 3
**Presentation:** 3
**Significance:** 2
**Originality:** 2
**Overall Recommendation:** 4
**Confidence:** 3

**Summary:**

The manuscript identifies "epistemic miscalibration in planning" as a distinct failure mode in LLM multi-agent systems. This occurs when agents confidently formulate plans that are logically executable but objectively infeasible due to misjudged knowledge limits. To mitigate this, the authors introduce the Epistemic Planning Calibration Agentic Workflow (EPC-AW). EPC-AW features Information-consistency-based Plan Selection (IPS) to favor plans evaluated consistently across agents with varied information contexts. It also incorporates Consistency-guided Epistemic State Refinement (CESR) to convert cross-round consistency signals into persistent planning constraints. Evaluation across six benchmarks demonstrates an average absolute accuracy improvement of 9.75%.

**Compliance With Llm Reviewing Policy:**

Affirmed.

**Key Questions For Authors:**

I have the following two questions for the authors:
- How does EPC-AW behave if the foundational LLM (Qwen3-Coder-30B) exhibits homogeneous overconfidence that overrides the artificially induced heterogeneous information states?
- Can you provide concrete examples of the lightweight epistemic constraints generated by the Diagnoser to better illustrate the CESR process?

**Limitations:**

Yes

**Strengths And Weaknesses:**

The paper has the following strengths and weaknesses:

**Strengths:**
- Isolating epistemic miscalibration from standard execution faults is a valuable conceptual contribution to multi-agent reliability.
- The IPS mechanism elegantly leverages cross-agent stability as a proxy for plan robustness without requiring external ground-truth validation.
- The empirical evaluation is thorough, utilizing six distinct datasets that balance closed-world reasoning and open-world search demands.

**Weaknesses:**
- The core assumption of IPS is that heterogeneous information states will expose miscalibration; however, shared systematic biases across identical backbone models might still lead to false consensus.
- The methodology introduces a noticeable computational burden, requiring six additional LLM evaluation calls per planning step. Although the authors note that these calls can be parallelized, the approach increases inference cost and may affect scalability.
- External tool latencies, such as the 80 seconds required for Google Search, pose practical limitations for time-sensitive deployments.
- The empirical evaluation is limited by its exclusive reliance on the Qwen3-Coder-30B backbone, leaving it unclear if the observed improvements generalize to other model architectures with different calibration profiles.

---

> ### Author Rebuttal · Authors · 2026-03-30
>
> We sincerely thank the reviewer for the insightful and constructive feedback. Below we address each concern in detail.
>
> ---
>
> ### **1. IPS under Shared Bias(W1/Q1)**
>
> Thank you for raising this concern. While shared biases in identical backbone models can cause false consensus, IPS mitigates this in two ways.
> First, IPS avoids reliance on consensus, and instead **evaluates the stability of plan quality across agents with heterogeneous information**. Second, IPS actively enhances epistemic diversity by **dynamically increasing the heterogeneity of agents’ information**. Specifically, agents are exposed to distinct historical signals (e.g., the Planner prioritizes failed executions, while the Diagnoser accumulates successful evidence), resulting in diverse informational contexts. This design ensures that even under homogeneous overconfidence, agents maintain divergent evaluations.
>
> Moreover, we compare IPS with a consensus-based baseline across tasks of varying difficulty (Sec. 5.4). Results show that **IPS outperforms a consensus-based baseline on complex reasoning tasks (+10.33% on HotpotQA, +3.67% on GAIA), highlighting its robustness.**
>
> ---
>
> ### **2. Computational Overhead and Tool Latency (W2/W3)**
>
> We thank the reviewer for raising concerns on efficiency and practicality.
> IPS introduces additional LLM evaluation calls, leading to a modest increase in time and token cost. However, this overhead is **a constant-factor addition** and does not change the system’s asymptotic complexity.
>
> Moreover, from a systems perspective, the overall wall-clock time is **dominated by external tool usage** rather than LLM inference. In realistic multi-hop pipelines (e.g., search → browse → summarize), a single Google Search call takes ~80s and Wikipedia retrieval ~40s, while each LLM call takes only ~3s (30B model). Therefore, the added evaluation calls contribute only marginally to end-to-end latency.
>
> Importantly, IPS does not introduce additional tool interactions, but only lightweight LLM-based calibration. Thus, it does not exacerbate the primary latency bottleneck, while significantly improving decision quality (+9.75% on average). **This leads to a favorable cost–performance trade-off, particularly in complex tasks where incorrect plans are costly and difficult to recover from**.
>
> Regarding **tool latency**, our implementation follows realistic multi-hop retrieval pipelines, like the Wikipedia tool used in the base AgentFlow setting (e.g., search → browse → summarize), where such sequential tool interactions are intrinsic to external evidence acquisition. IPS operates within this pipeline without increasing tool usage, ensuring that the dominant latency source remains unchanged.
>
> We have included these discussions and clarifications regarding computational overhead and tool latency in the revision.
>
> ---
>
> ### **3. Generalization on Different LLMs (W4)**
>
> We appreciate this important concern. We extend experiments to additional backbones, including **Qwen3-14B** and **DeepSeek-R1-32B**, on two representative datasets. Results show **consistent improvements across architectures**, indicating that EPC-AW improves epistemic calibration rather than relying on backbone-specific properties.
>
> | Method     | Qwen-14B (Bamboogle) | Qwen-14B (HotpotQA) | DeepSeek (Bamboogle) | DeepSeek (HotpotQA) |
> |---|----|---|----|---|
> | Non | 45.60| 42.00 | 48.80| 49.00 |
> | Retry      | 48.00| 49.00 | 51.20| 53.00 |
> | Rollback   | 48.80| 51.00 | 53.60| 57.00 |
> | EPC-AW     | 56.80| 58.00 | 62.40| 63.00 |
>
> ---
>
> ### **4. Examples of CESR Constraints (Q2)**
>
> We thank the reviewer for the suggestion. We provide two examples illustrating how CESR generates epistemic constraints:
>
> **1) Tool Misuse** (bamboogle 8.json)
> The Planner assumes a ScienceDirect URL provides primary evidence, but the Executor returns no content, revealing a tool-capability mismatch. CESR redirects the plan to feasible alternatives (e.g., Google Search).
>
> **Generated constraint:**
> *If a tool or URL fails to provide the required evidence, stop using it and do not assume alternatives will automatically succeed; instead, revise the plan toward achievable, verifiable sources.*
>
> **2) Goal Mis-specification** (bamboogle 11.json)
> The Planner assumes a complete list of Theranos whistleblowers can be obtained via repeated Google searches, but exhaustive information is unavailable. CESR refines the goal to focus on verifiable cases from available evidence.
>
> **Generated constraint:**
> *If a task depends on inaccessible or authoritative sources, avoid assuming full retrieval is possible; revise the goal to align with verifiable evidence.*
>
> We have included more detailed case studies in the revision.

---

> > ### Author Rebuttal · Reviewer_tTHj · 2026-04-02
> >
> > Thank you for the detailed rebuttal. I have read it carefully and want to share where things stand from my perspective.
> >
> > On efficiency (W2/W3), the argument that tool latency dominates runtime does not hold across the full benchmark set. Bamboogle relies minimally on external search and MedQA is entirely closed-book. The rebuttal does not address these cases separately, so the efficiency justification is at best benchmark-specific rather than a general property of the system.
> >
> > On generalization (W4), the added experiments cover two of six datasets. This is insufficient to support a broad claim that EPC-AW generalizes across model architectures.
> >
> > On W1, the rebuttal does not resolve the core concern. Divergent memory contents do not guarantee divergent evaluations if the backbone model's overconfidence is insensitive to information context. This is a fundamental question about whether the mechanism works for the reasons claimed.

---

> > > ### Author Response · Authors · 2026-04-06
> > >
> > > We sincerely thank the reviewer for the detailed and thoughtful feedback.
> > >
> > > ---
> > >
> > > ### 1.  On Efficiency
> > >
> > > We agree that tool latency varies across benchmarks, and our previous explanation may appear benchmark-specific. We clarify that our efficiency claim does not depend on any particular tool assumption. Even in settings with minimal or no external tool usage, IPS introduces only a **constant-factor overhead**.
> > >
> > > In the base workflow, each round consists of one planning call, one tool execution, and two diagnosis-related LLM calls.
> > > Due to the additional evaluation calls in IPS can be parallelized using modern LLM serving frameworks (e.g., vLLM), these calls can be executed concurrently with **latency comparable to a single LLM call**. Moreover, tool execution itself is often non-trivial in latency. Beyond relatively slow tools such as web search, even lightweight components (e.g., base LLM generators, Python coder) typically incur **3–5 seconds per call**. **Under the most lightweight tool setting, IPS increases total runtime by only about 25%**.
> > >
> > > Overall, this moderate overhead is accompanied by a **+9.75% average performance improvement**, resulting in a favorable cost–performance trade-off, particularly in complex tasks where incorrect plans are costly and difficult to recover from.
> > >
> > > ---
> > >
> > > ### 2. On Generalization
> > >
> > > We appreciate the reviewer’s concern regarding the limited number of additional datasets.
> > >
> > > Our goal is to demonstrate that EPC-AW remains effective under **both model scale and architectural variation**. To this end, we selected two representative settings with different LLM backbones: **Qwen3-14B** (smaller-scale model) and **DeepSeek-R1-32B** (reasoning-oriented architecture). This setup allows us to evaluate generalization across both parameter scale and reasoning paradigms.
> > >
> > > We agree that broader empirical coverage would further strengthen the claim. The current results already demonstrate consistent improvements across different settings, and additional datasets will be included in the revision to further validate generality.
> > >
> > > ---
> > >
> > > ### 3. Mechanism Validity and Overconfidence Concern
> > >
> > > We appreciate the reviewer’s concern and acknowledge the possibility that divergent memory may not lead to divergent evaluations, if the backbone model’s overconfidence is entirely insensitive to context. However, our design in IPS **actively promotes evaluation diversity by dynamically increasing the heterogeneity of agents’ contextual information**, and we do **not observe this failure mode empirically**.
> > >
> > > Specifically, we provide **quantitative evidence at both the evaluation level and the IPS decision-signal level**, showing that contextual differences do lead to evaluation divergence.
> > >
> > >
> > > **1) Evaluation disagreement on the same plan.**
> > > We measure variance across agents for the same plan, averaged over all plans and samples:
> > >
> > > - **HotpotQA:** Step 1: 0.07 → Step 2: 0.16 → Step 5: 0.31
> > > - **Bamboogle:** Step 1: 0.05 → Step 2: 0.12 → Step 5: 0.28
> > >
> > > This yields a consistent ~4-5× increase, indicating **agent evaluations increasingly diverge due to contextual differences**.
> > >
> > >
> > > **2) IPS leverages a more discriminative signal.**
> > >
> > > **Importantly, IPS selects plans via information-consistency, which measures the stability of plan evaluations across agents, rather than relying on raw evaluation scores.** Its variance across plans also increases:
> > >
> > > - **HotpotQA:** Step 1: ≈0.002 → Step 2: ≈0.020 → Step 5: ≈0.045
> > > - **Bamboogle:** Step 1: ≈0.002 → Step 2: ≈0.028 → Step 5: ≈0.064
> > >
> > > This ≈20–30× increase shows that **the IPS decision signal becomes increasingly discriminative as contextual divergence grows**, enabling more reliable plan selection.

---

### Decision · Program_Chairs · 2026-04-30

**Decision:**

Accept (regular)

**Comment:**

The four reviewers have consistently agreed that this is a sufficiently strong, sound and ICML-relevant paper to deserve acceptance (with 2 Accept and 2 Weak Accept recommendations).

We are grateful to the reviewers and authors for their constructive comments and focused responses during the review process. The discussions on their own would be interesting and relevant to the ICML community, who would most certainly engage well with this research work and paper during the conference.

We encourage the authors to carefully consider and integrate the changes and responses in the final version of the paper.